# Community-Based Watershed Change: A Case Study in Eastern Congo

**Scott Sabin [1], Birori Dieudonne [2], John Mitchell [1], Jared White [1], Corey Chin [1] and Robert Morikawa [1,\*]**

1   Plant with Purpose, 4747 Morena Blvd, San Diego, CA 92117, USA; scott@plantwithpurpose.org (S.S.); john@plantwithpurpose.org (J.M.); jared@plantwithpurpose.org (J.W.); corey@plantwithpurpose.org (C.C.)
2   Ebenezer Ministries International, Uvira, 0970 South Kivu, Democratic Republic of the Congo; dieudonnebirori@gmail.com
\*   Correspondence: robertmorikawa@gmail.com

**Abstract:** Conflict and environmental degradation in the Democratic Republic of the Congo are interrelated and complex. The authors conducted a case study of a community-based environmental restoration project in Eastern Congo and provide early results which suggest a link between community environmental action and multidimensional outcomes such as peace and reconciliation. The project examined in this study is based on a framework (Theory of Change) which networks communities through autonomous savings groups, churches, mosques, schools, and a community leadership network with the goal of catalyzing sustainable farming, reforestation, and community forest management. The primary project input was training, and the resulting voluntary community action included tree planting and the management of common forest areas. A mixed-methods approach was used to evaluate project results comparing two watersheds, and included a difference in differences analysis, participatory workshops, remote sensing analysis, and community activity reports. Positive change was observed in the treatment watershed in terms of ecosystem health and household economic condition. Results suggest a possible influence on peace conditions which, while fragile, offers hope for continued restorative action by communities. This study provides evidence that a community-based approach to environmental restoration may have a positive influence on multidimensional issues such as forests, watershed health, economic well-being, and peace.

**Keywords:** watersheds; community-based restoration; Village Savings and Loans Associations; ecosystem health; sustainable farming; agroecology; peace and reconciliation; community forest management

## 1. Introduction

The Eastern Democratic Republic of the Congo (DRC) is a region with vast potential in both natural and human resources. However, long-term conflict in the region has resulted in accelerated forest loss [1–3]. In a post-conflict situation such as the DRC, the peace process tends to focus on the logistical and institutional aspects of peace, such as negotiations between political groups and reintegrating armed groups into civil society. This process does not typically take natural resource management into account, and can contribute to environmental degradation as communities made desperate by conflict exploit forest resources [4]. The relationship of rural communities in the DRC with forests and natural resources is complex [5], and conflict in particular can affect wildlife and natural resources through multiple pathways [6], such as a direct impact on wildlife species (e.g., the mountain gorilla (*Gorilla beringei*) in Rwanda [7]), to institutional impacts such as reducing enforcement effectiveness [8]. Addressing issues of conflict, peace, and deforestation requires an understanding

of how complex pathways connect and interact [6]. The purpose of this paper is to present some early results of a case study which may contribute to the understanding of how a community-based approach to forest management can contribute to these complex and multidimensional pathways of change—changes such as increasing forest cover, improving human well-being, and contributing to peace and reconciliation in a post-conflict zone.

Worldwide, the poor have been blamed for deforestation, because of the expansion of small-scale farming, the cutting and making of charcoal, and rotational farming (i.e., "slash and burn" agriculture). In spite of this, poverty can often be associated with less deforestation [9], although causes of this association have not necessarily been elucidated. An extensive review of the literature by Adams et al. shows that there is a considerable body of evidence suggesting that biodiversity conservation needs to consider the factor of human poverty [10]. Nevertheless, community-based approaches to conservation are controversial, and have shown mixed results [10–12]. Berkes asserts that it is important to consider cases where community-based approaches have worked, and where they have not [11]. Conservation initiatives have shown positive outcomes when there is an emphasis on sustainable resource management and community involvement [13], and both community livelihoods and forests benefit [14,15]. A community-based approach to conservation has been demonstrated to be successful in the DRC [16], and is seen by some authors as an important component of forest planning policy in the DRC [17]. Elsewhere, authors discuss how community networks have been demonstrated to be an effective way of managing forests at the watershed level in Northern Thailand [18,19].

Reviews of the literature point to evidence showing how community networks and local involvement can enhance forest management by devolving the decision-making process, integrating local knowledge, and increasing secure tenure [11,15,20]. For example, a long-term comparison in India showed that community-managed forests are at least as effective and less costly than state-managed forests [21]. Community forestry is a more recent development in Africa, but case studies from Tanzania, where community forestry has been established since the early 1990s, show improving forest conditions where local actors are empowered [22]. There is less evidence of the effect of community forestry in Congo [23], but studies from other regions such as Nepal have looked at the connection between increased conflict resilience and local empowerment through community forest action [24,25].

The initiative examined by the authors in this case study placed priority on catalyzing community action rather than direct inputs. This means promoting behavior change among community members so that, for example, farmers themselves are voluntarily planting trees, rather than a project planting trees on behalf of farmers or paying them to plant trees. The initiative in this case study employed several strategies toward this end.

One of these strategies was the networking and training of farmers to promote sustainable farming practices. Communities in the Eastern DRC where this study took place are at the nexus of forests and farmland. As a result, community forest restoration in this context is significantly impacted by trees on farmland. On the farmland side of the equation, social networks can help to catalyze community action by allowing the sharing of local germplasm, sharing knowledge of adaptive farming strategies, and the rapid scaling up of resilient techniques, and have been shown to be closely tied to building agroecological resilience [26]. Farmer experimentation and the adoption of sustainable farming practices such as agroforestry is promoted through social learning and a participatory approach [27,28]. The adoption of environmentally friendly practices is positively influenced by group participation and learning [29–31].

A second strategy used by the project to catalyze community networks and community action was through the promotion of savings groups or Village Savings and Loans Associations (VSLAs) [32]. Lack of infrastructure and financial services makes microsavings particularly appropriate for rural areas [33,34]. Extensive study in sub-Saharan Africa has shown that the VSLA methodology can have a positive effect on poverty [35], although outcomes vary from study to study. For example, one randomized control trial (RCT) in Ghana, Malawi, and Uganda showed a positive impact of VSLAs on women's empowerment [36], while another RCT in Mali showed no influence on downstream

effects such as women's decision-making power [37]. These varying and sometimes conflicting results underscore the caution that should be exercised when trying to extrapolate lessons learned from complex environments, including the current case study. Positive economic outcomes of savings groups reported by various studies include improved food security, increased household expenditure, improved children's health, and education [38,39]. A study specific to the DRC showed that VSLAs can contribute to improving financial management skills and the financial transparency of local institutions in mining regions, which has helped address issues of corruption [40]. Fewer studies have reported or looked at the environmental impacts of savings groups, but several projects have intentionally integrated VSLAs as a component of an environmental initiative [41–43]. One project in Kenya has shown that farmers involved in VSLAs are better able to spread risk by making better decisions about climate-smart agriculture technologies such as crop diversification and agroforestry [44].

Making the link between community forest action and improved forest condition is not easy. Numerous analyses have shown that the relationship is complex, and results are sometimes conflicting [11,14,45]. Making the link to other multidimensional issues such as improved human well-being or peace and reconciliation is equally or more challenging [14,46]. Very few studies among the hundreds conducted provide quantitative evidence of these complex links [14]. Nevertheless, a multidisciplinary approach to issues of forest degradation and conservation is being recommended both globally and for the DRC in particular [11,17]. This case study provides quantitative and qualitative evidence of an intervention which shows hopeful, albeit early, signs of catalyzing community action to restore forest ecosystems, and shows promising links to multidimensional outcomes such as economic well-being and peace and reconciliation.

## 2. Materials and Methods

We report on a community-based initiative in Uvira Territory of South Kivu, in the Eastern Democratic Republic of the Congo. A Theory of Change (TOC) framework outlines a watershed-level restoration model which integrates VSLAs as a platform as well as existing institutions such as churches, mosques, and schools for strengthening household economic condition and catalyzing downstream impacts such as environmental restoration. A conceptual framework such as a TOC provides a way to articulate goals, assumptions, and metrics [47]. In a complex context such as a watershed- or landscape-level community effort, having such a framework with which to iteratively assess progress towards short-, mid-, and long-term goals becomes critical [48]. A mixed-methods approach to analysis was employed incorporating a difference in differences (DID) analysis of two neighboring watersheds. Authors were involved in conceptualization and implementation of both the project and the study, although external surveyors were hired to conduct interviews for the household survey.

The study area (see Figure 1) was located in South Kivu province of the DRC in Uvira territory. According to Demographic Health Survey (DHS) data from 2013 [49], in South Kivu 70.7% of women and 45.6% of men work in agriculture, 59.8% of women are literate, and 26% of children are underweight. Two watersheds in Uvira—Kakumba and Kambekulu—were selected for the study based on similarity in size, ecology, and demographics, in addition to being close to the partner organization office. Some basic characteristics of these watersheds are shown in Table 1.

**Table 1.** Characteristics of the two watersheds at the study site.

| Watershed | Area sq km | Population [50] | percent Tree cover 2010 [51] | Designation |
|-----------|-----------|-----------------|------------------------------|-------------|
| Kakumba | 37 | 13,917 | 22.6% | treatment |
| Kambekulu | 54 | 16,338 | 27.5% | comparison |

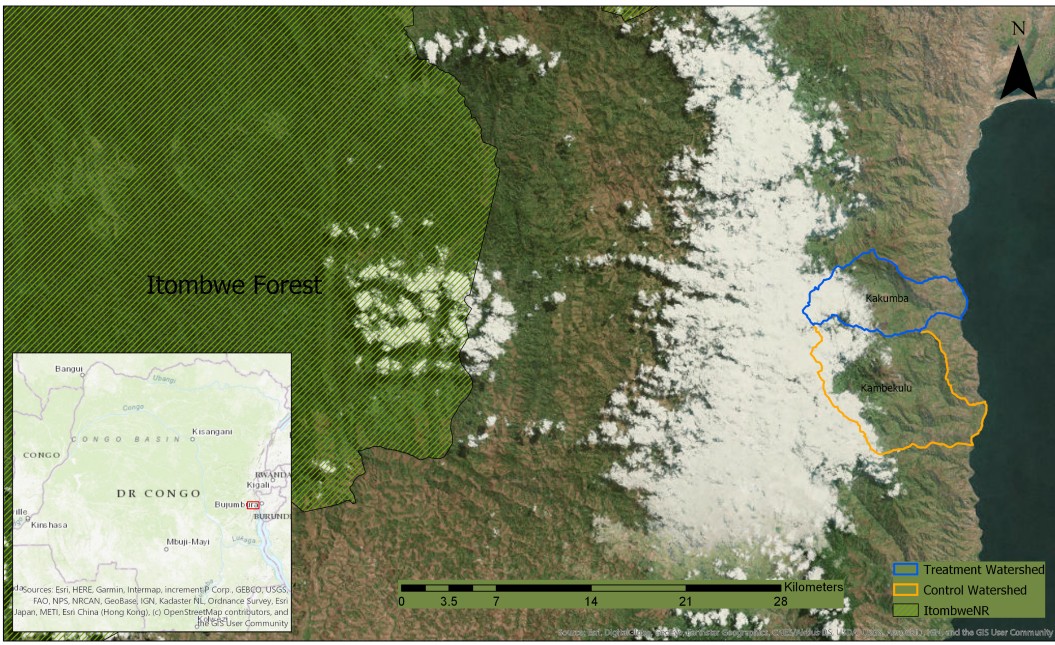

**Figure 1.** Map of the study watersheds.

Both watersheds at the study site are primarily agricultural, but have significant forest cover—especially in the upper watershed areas. The study site is adjacent to the Itombwe forest—a globally recognized area of biodiversity [52].

Beginning in 2015, a community-based watershed project was implemented in the study area through a collaboration between two organizations: Ebenezer Ministries International (EMI), the local implementer; and Plant With Purpose [53], the international partner. One of the study watersheds, Kakumba, was randomly selected as the treatment. Note that while selection of one watershed over the other at the study site was random, participation in the program within the selected target watershed (Kakumba) was not random, and was based on household interest. The treatment involved farm communities in the watershed working with the implementing partner to establish autonomous savings groups and the promotion of community-based environmental restoration activities—both on- and off-farm—with the savings groups, serving as a platform for those complementary activities. Savings groups were established according to the standard VSLA methodology [32]. Promotion of environmental restoration activities included the use of a sustainable farming training curriculum, farmer field schools, and the establishment of a community leadership network—consisting of government representatives, police, religious, and traditional leaders—to catalyze environmental awareness. The treatment methodology was based on an organizational Theory of Change (TOC) which served as a framework for the treatment as well as the evaluation protocol. The relevant environmental restoration nodes of the TOC map are shown in Figure 2. The TOC is intended to illustrate a logical progression from short-term outputs to long-term impacts. In particular, it is worth noting that "stakeholders are catalyzed and networked" through the formation of savings groups and working with other local institutions such as churches, mosques, and schools; the expected outcome of this networking is that communities "value and protect their natural environment". At each node in the TOC pathway, from outputs, through outcomes, to impact, indicators were developed and then tested using all available evidence from those indicators. A range of indicators and indicator formats were developed, including qualitative indicators used in participatory workshops, activity indicators collected on a quarterly basis, quantitative indicators collected in a formal household survey, and remote sensing data in the form of the Normalized Difference Vegetation Index (NDVI). Testing of the TOC pathway using available indicators is further explained in the Discussion.

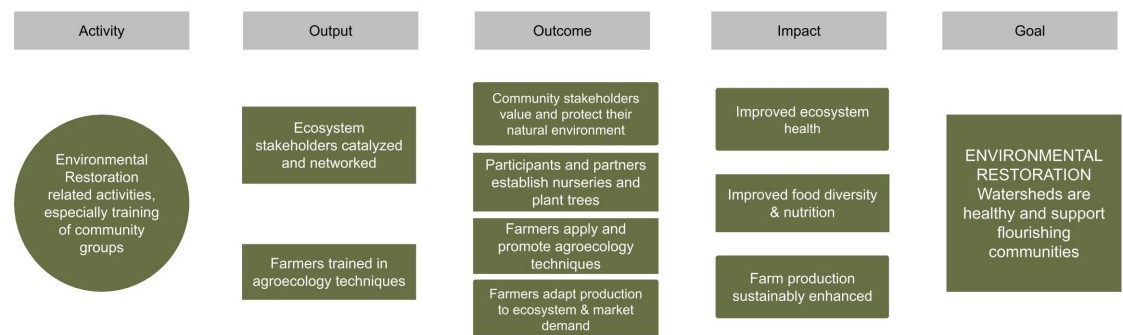

**Figure 2.** Theory of Change—Environmental restoration nodes.

Phase one of the study was conducted in January 2015 in order to collect baseline data using participatory workshops and a household survey of 96 randomly selected households from the Kakumba watershed and 87 randomly selected households from the Kambekulu watershed. Phase two of the study was conducted at the end of the pilot project in May 2017, also using participatory workshops (see Appendix B) and a household survey of 160 randomly selected households from each watershed for a total of 320 households sampled (see Appendix A for more details on the indicators collected). Total sample size was based on the total population in the watersheds with the aim to obtain a confidence interval of 7.5% on estimates of the mean. The 2017 sample size was increased so that a similar level of confidence could be obtained when data were disaggregated by factors such as watershed, gender, and participation. In order to minimize sample bias, complete household lists of all households in each village in both the treatment and control watersheds were collected in 2017. Households to be surveyed were then randomly selected from those complete lists to generate the 320 sampled households. This was done to reduce the risk of favoring participant households in the treatment watershed over non-participant households. In other words, the sample in the treatment watershed was made up of households who participated in the program as well as households who did not participate. Quantitative results were assessed using a difference in differences analysis [54]. Difference in differences analysis compares the change in condition of two groups, the treatment and comparison (control) over a period of time—essentially a statistical "before and after" assessment. Throughout the project period, vegetation cover change was monitored using the NDVI product of MODIS MOD13A3 data [55] and ESRI ArcGIS Pro software [56]. In addition, treatment communities reported every 3 months on short-term metrics such as trees planted and savings group activities. These short-term metrics were used to corroborate the longer-term metrics measured in the household survey and participatory workshops. All metrics were examined to test the Theory of Change at each node from output to impact level using a contribution analysis approach [57]. All statistical data were analyzed using the R statistical platform [58]. All sources of information, the participatory workshop discussions, the DID of the household survey results, the vegetation cover analysis, and community activity reports were analyzed using a mixed-methods approach in order to try to develop a full picture of treatment effect (if any).

## 3. Results

Over the study period (January 2015 to May 2017), 21 VSLA groups were established in the treatment watershed, representing approximately 30% of the watershed population. Table 2 summarizes some key activities of the savings groups at the end of the study period (June 2017). Average value of savings held by the groups and group member attendance were stable or increased over the study period, as shown in Figures 3 and 4.

**Table 2.** Summary of savings group activities, treatment watershed. VSLA: Village Savings and Loans Association.

|  | **June 2017** |
|---|---|
| VSLA groups | 21 |
| VSLA members | 609 |
| VSLA member equity (USD) | 11,160 |
| Trees planted | 165,000 |

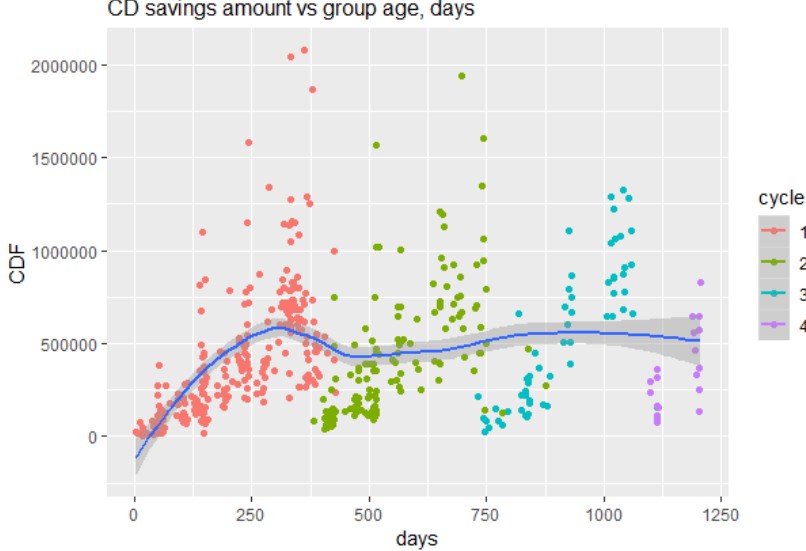

**Figure 3.** Trend in average savings per group (Congolese Francs).

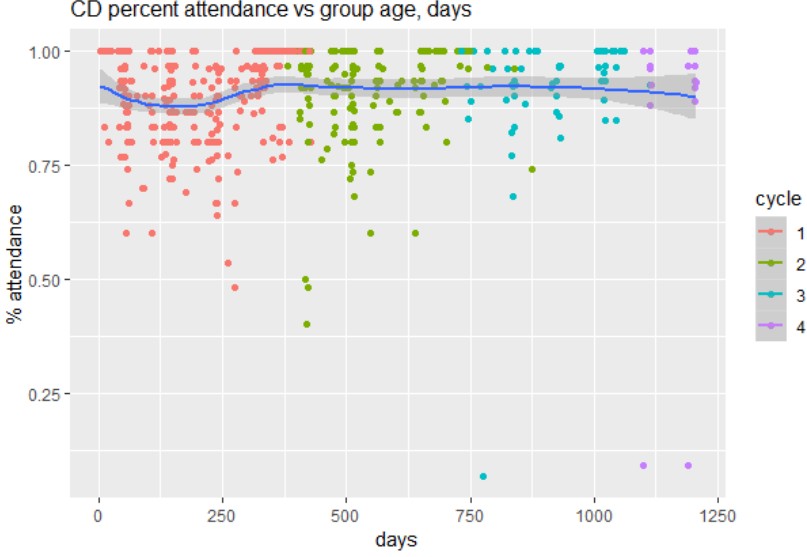

**Figure 4.** Trend in attendance per group.

The effect of savings groups was also noted in participatory workshop results. Workshop participants in the treatment watershed, where VSLA groups were established, identified an increased availability of capital as a result of savings group activity as one of the most significant changes in the watershed, while the workshop participants in the control watershed (where no VSLA groups were established) identified a lack of capital and increasing poverty as a significant change over the study period. In the VSLA methodology, self-selected groups saved regularly and made loans to group members, establishing their own bylaws, leadership, and loan interest rates. A high priority was placed

on transparency, which promotes confidence among members. At the end of a savings cycle (typically 12 months), group members divided accumulated savings capital and interest earned from loans, each member receiving an amount proportionate to their investment. Most groups agreed to begin another savings cycle, and it was common for members to agree to invest a portion of their savings from the previous cycle, so that more capital was available for credit purposes. Growth of the savings amount as shown in Figure 3 was partly a result of this group's decision to reinvest in the subsequent cycle. Group members typically used both credit from the group as well as their accumulated savings to invest in small business, education, or to diversify and protect their farms.

Twenty-one indicators were analyzed using difference in differences analysis for randomly selected households in the treatment and control watersheds. Fifteen of the twenty-one indicators showed improvement for the treatment watershed relative to the control. A detailed list of indicators can be found in Table 3, suggesting improvement in both economic and environmental conditions.

**Table 3.** Difference in differences analysis, all changes tested at $p = 0.05$ confidence level.

| Indicator | Improvement | No Change | Decline | $p$-Value |
|---|:---:|:---:|:---:|---|
| Savings habit | X | | | $7.98 \times 10^{-16}$ |
| Savings amount per household | X | | | $6.99 \times 10^{-4}$ |
| Cell phone ownership | X | | | $1.01 \times 10^{-8}$ |
| Rooms per household | X | | | $3.57 \times 10^{-5}$ |
| Households with girls in secondary school | X | | | 0.00494 |
| Income diversity | X | | | $3.30 \times 10^{-16}$ |
| Crop diversity | X | | | $9.67 \times 10^{-9}$ |
| Percent of land protected per household | X | | | $1.36 \times 10^{-10}$ |
| Soil quality | X | | | $2.00 \times 10^{-16}$ |
| Sustainable farming technique diversity | X | | | $2.00 \times 10^{-16}$ |
| Meals per day | X | | | $2.00 \times 10^{-16}$ |
| Nutrition diversity | X | | | $1.08 \times 10^{-7}$ |
| Trees planted per household/year | X | | | $3.44 \times 10^{-7}$ |
| Households planting native tree species | X | | | 0.00827 |
| Households donating to local church/mosque | X | | | 0.000584 |
| Incidence of dirt floor in household | X | | | 0.992 |
| Households owning land | | X | | 0.1979 |
| Households reducing meal portions in past 30 days | | X | | 0.1842 |
| Average time to and from drinking water | | X | | 0.0724 |
| Households selling crops | | X | | 0.86889 |
| Amount of land owned per household | | | X | 0.01861 |

The household survey and the DID analysis confirmed an increase in the average number of trees planted per household in the treatment watershed (Figure 5). This was also supported by the participatory workshop results, where "an increase in trees" and "decreased erosion" were noted as two of the significant changes in the treatment watershed. Households in the treatment watershed also noted a 75% decrease in the practice of burning fields during land preparation—a practice which has historically contributed to the degradation of soil quality and tree cover loss. In the case of all environmental restoration activities (i.e., tree planting, sustainable farming techniques, reduced burning, community forest management), these are activities in which communities engage voluntarily. These are "outputs" in the TOC, and the principle input from the project is training for VSLA formation and environmentally friendly farming practices. Figure 6 shows one of several community forest areas which is being actively managed for conservation as well as production.

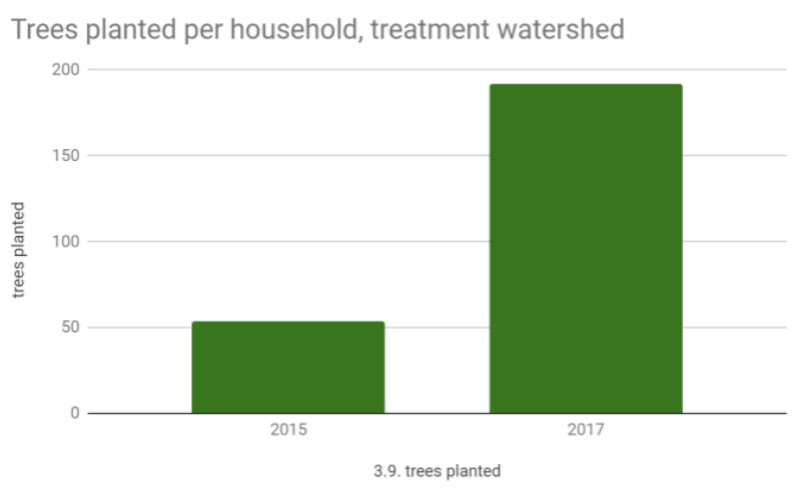

**Figure 5.** Trees planted per household—treatment watershed.

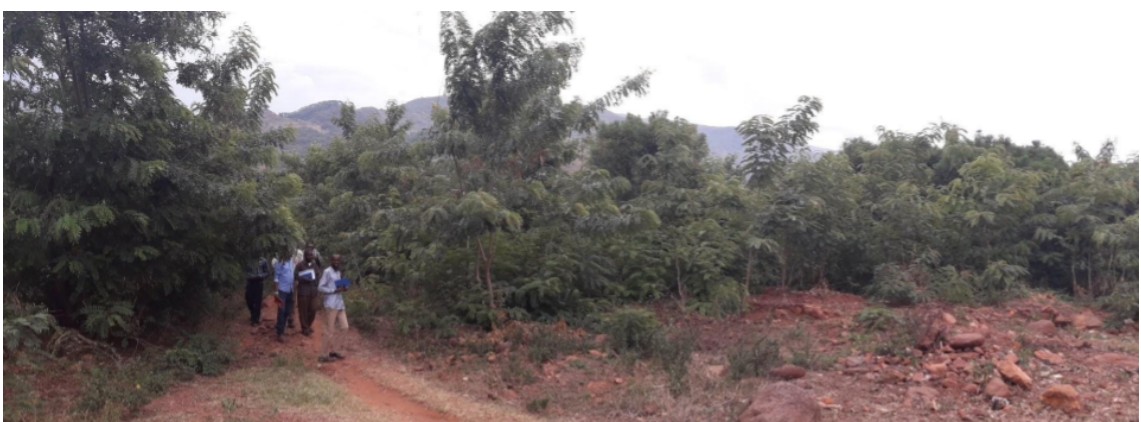

**Figure 6.** Community forest area—treatment watershed.

In both watersheds, NDVI analysis showed that vegetation cover increased over the study period, although no statistically significant difference between the treatment and control watersheds was detected. While NDVI change analysis has been used successfully by the implementing organization in the past [43], significant cloud cover in the study area limited the available data, resulting in a more challenging analysis. Furthermore, there is often a lag between community restoration activities and measurable vegetation change using NDVI data. In this case, the 2-year study period may not have been long enough to detect vegetation change using remote sensing methods.

A marked difference in women involved in leadership roles was also observed between treatment and control watersheds. Figure 7 shows that women in the treatment watershed were nearly four times more frequently involved in leadership roles.

Community leadership has also had a significant role in promoting tree planting and community forest management. Local leaders as well as the community network have promoted tree planting in and around the forest in the upper watershed, and have enforced local regulations. This resulted, for example, in the imposition and collection of fines for illegal tree cutting—a practice which has been neglected in the past.

Perhaps some of the most significant and unexpected changes noted by workshop participants related to the theme of peace and reconciliation. Community and religious leaders from the treatment watershed have observed several indications of reduced conflict, including:

- Increased cooperation within and between communities;
- Removal of trade barriers—specifically illegal roadblocks and tolls—between communities;
- A decrease in local court cases;

- Community leaders successfully discouraging youth from joining armed groups.

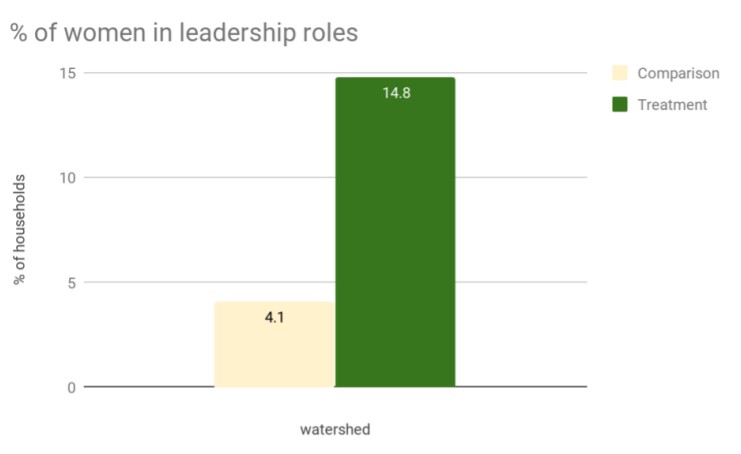

**Figure 7.** Percent of households with women in leadership roles, 2017.

## 4. Discussion

Results of the study were assessed within the framework of the project Theory of Change, and evidence was considered at each node of the TOC map, adopting a contribution analysis methodology [57]. Figure 8 shows a partial contribution analysis of three nodes of the TOC, aggregating data from the DID analysis, the participatory workshops, remote sensing analysis, and community activity reports. Considered in aggregate, a strong case can be made that:

- Savings groups were established and became stronger over the study period;
- Groups were a catalyst for tree planting, more sustainable farming, and contributed to less burning;
- Group/community action resulted in improved ecosystem health in the treatment watershed as evidenced by more trees, improved community forest management, and healthier farms.

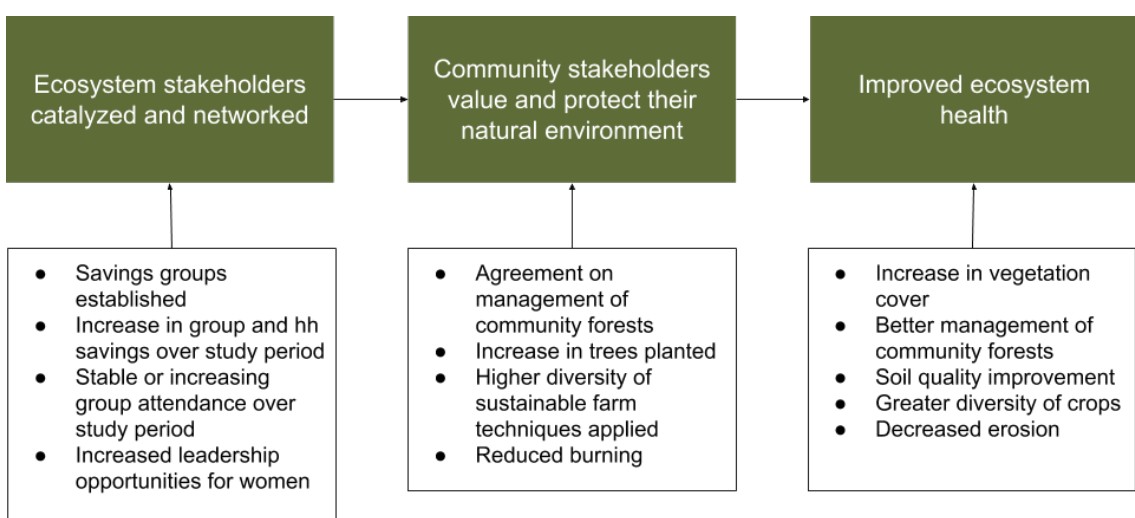

**Figure 8.** Partial contribution analysis of Theory of Change (TOC).

The burning of farm fields in preparation for planting is a long-standing practice in rural communities in the Eastern Congo, and the large reduction in the use of fire as a farming practice in the target watershed represents a significant behavior change. As noted elsewhere in this paper,

these are early data, but if this change in farming practices is sustained it could potentially result in a positive influence on tree cover and soil health both on and off farm land.

Action by local leadership to promote forest protection such as through increased enforcement of local environmental regulations is another behavioral change noted at the watershed level (rather than at the farm or household level, as is the case with burning). Others have observed that stable and empowered leadership is critical to the sustainability of community forest practice [15,25]. Effects on leadership are certainly less tangible, and this study does not allow claims about causal relationships with the target intervention. Nevertheless, increased environmental restoration action by leaders and increased leadership opportunity is one of the outcomes that we propose has potential to contribute to sustainable environmental change in the target watershed.

Improved signs of peace and reconciliation as noted by workshop participants are worth highlighting. The analysis of peace and conflict is complex [5,6,59], and the connection between "ecosystem stakeholders being networked" and peace should be considered with caution. Nevertheless, workshop participants observed that increased cooperation, as well as a sense of ownership for the resulting economic and environmental gains seen in the watershed, resulted in healing of relationships and strengthened community cohesion. Community and religious leaders reported working together to actively discourage youth from joining armed groups, because of the common feeling that association with armed groups through recruitment may compromise the gains that are perceived to be taking place. Armed groups have been a significant source of instability, and although the situation in Eastern Congo is widely recognized as fragile, these early signs are hopeful.

This comparative study of two watersheds in the Eastern Congo presents evidence of watershed change catalyzed by savings group and community action through existing local institutions such as churches, mosques, and schools. In addition to Savings and Loan activities, group members took restorative measures on their farms and in commons spaces, resulting in greater vegetative cover, improved community forest management, improved soil quality, and reduced erosion. The study also observed a concomitant improvement in household economic condition in the treatment watershed—an effect which is not frequently reported in community forestry initiatives [45].

Less-tangible outcomes were observed by communities, including more leadership opportunities for women, leadership action to protect community forests, strengthening of community cohesion, and more peaceful relationships within the watershed. While more difficult to conclusively demonstrate, the potential strengthening of peace is of particular significance given the history of conflict in the region. While hopeful, these gains are also fragile, and it has been asserted that conservation efforts in the context of conflict found in the DRC require a long-term commitment [60].

Both the role of community-based restoration as well as the effect of community based restoration are subjects of considerable debate [10–15,20,22,23]. Very few studies of community forest action present quantitative evidence of multidimensional change such as improved livelihoods, increased leadership opportunities for women, and positive influence on peace conditions. This case study does provide such evidence using a statistically valid method (difference in differences analysis) as well as multiple sources of evidence (participatory workshops, remote sensing data, community activity records). We as authors and practitioners are aware of the limitations of a short-term study such as this, as well as the danger of linking early results to long-term impact. Nevertheless, the outcomes observed in this study give us a great deal of hope, and we would argue warrant a closer look in informing the ongoing discussion of the role and value of community forest action.

## 5. Conclusions

Although it was conducted in the short term (2.5 years), this study strongly suggests that the voluntary actions taken by farm communities in the treatment watershed may be linked to multidimensional outcomes. These outcomes include improved livelihood conditions, increased leadership opportunities for women, and increased community action leading to improved watershed health. The intervention itself is multidisciplinary in nature—an approach advocated by others as an

appropriate way to address forest management in the Congo basin [17]. The formation of savings groups helps to address basic economic issues, and gives farmers greater latitude to change farming and forestry practices. Training and networking gives community members options, allows the exchange of ideas, and promotes the collective management of farms and common forest areas. The outcomes resulting from these community actions support the results of other studies that show that catalyzing community involvement can be an effective alternative for promoting the improvement of watershed/ecosystem health and forest management [60,61].

As Berkes expresses so succinctly, "Asking whether community-based conservation works is the wrong question. Sometimes it does, sometimes it does not. Rather, it is more important to learn about the conditions under which it does or does not work" [11]. Given the massive economic, political, and ecological challenges faced by rural communities in the Eastern DRC, there is much to be learned about what works and what does not. Over a 2.5-year period, communities in the target watershed were witness to early but significant changes within that difficult context, corroborated by the findings presented here.

As the authors and practitioners involved in the intervention that was the subject of this study, we are cognizant of the risks in drawing conclusions about long-term impact from these early results. Nevertheless, we submit this as an example of a community-based initiative that is working, and we are hopeful about the long-term impacts, as are community members in the target watershed (see Figure 9). Long-term study is planned and lessons learned—both positive and negative—can contribute to understanding the role of community-based efforts in conserving forests and watersheds.

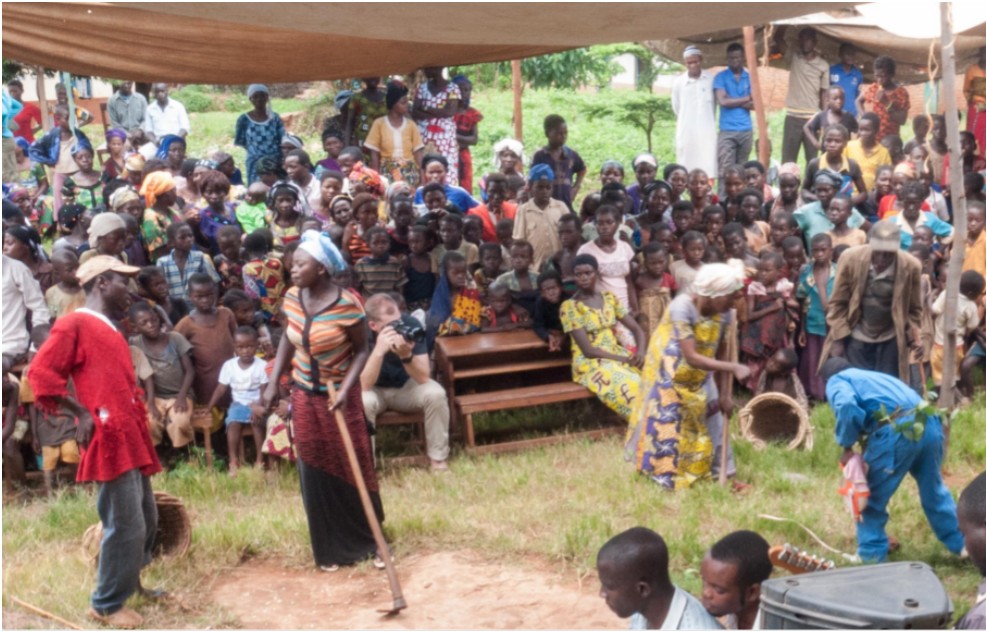

**Figure 9.** A play, written and performed by savings group members about the value of sustainable farming practices.

**Author Contributions:** Conceptualization, B.D., S.S., J.M., R.M.; methodology, B.D., R.M.; formal analysis, B.D., R.M.; data curation, C.C.; writing–original draft preparation, R.M.; writing–B.D., S.S., J.M., J.W., C.C.; visualization, C.C., R.M.; project administration, B.D., J.W.; funding acquisition, S.S.

**Funding:** This research received no external funding

**Acknowledgments:** Special recognition and gratitude to the thousands of farmers in eastern Congo who have dedicated their talents and energy to restoring their farms and their watersheds.

**Conflicts of Interest:** The authors are employees of Plant With Purpose and have been directly involved in design, implementation, monitoring and evaluating the project discussed. The authors have a strong interest in objectively understanding pilot project outcomes to determine appropriate modification, termination or continuation of the approach. The methods of evaluation are structured to give a balanced view. Involvement of different departments

reviewing results provide checks and balances within the organization. The results documented in this report have led to expanded operations and increased funding.

**Abbreviations**

The following abbreviations are used in this manuscript:

| | |
|---|---|
| DID | Difference in differences |
| DRC | Democratic Republic of the Congo |
| EMI | Ebenezer Ministries International |
| hh | household |
| MODIS | Moderate resolution imaging spectroradiometer |
| NDVI | Normalized difference vegetation index |
| RCT | Randomized control trial |
| Sq km | square kilometres |
| TOC | Theory of change |
| VSLA | Village Savings and Loans Associations |

**Appendix A**

Indicators collected and analyzed in DID analysis

1.  savings habit: frequency of households who are actively saving cash
2.  savings amount per hh: mean number of months that households estimate they have in emergency reserves
3.  cell phone ownership: frequency of households owning a cell phone
4.  rooms per hh: mean number of room in households
5.  hh with girls in secondary school: frequency of girls regularly attending secondary school (at least 10 day per month)
6.  income diversity: mean count of income sources per household
7.  crop diversity: mean count of crops harvested in the past 12 months per household
8.  percent of land protected per hh: mean percentage of land per farm protected with trees or soil conservation measures
9.  soil quality: mean farmer perception of soil quality on their farm, scale of 1–5
10. sustainable farming technique diversity: mean count of sustainable farming techniques applied per household
11. meals per day: mean number of meals per day per household
12. nutrition diversity: mean index score based on frequency of consuming 5 food categories per household
13. trees planted per hh/year: mean number of trees planted in the past 12 months per household
14. hh planting native tree species: frequency of households planting native tree species
15. hh donating to local church/mosque: frequency of households donating cash to a church or mosque
16. Incidence of dirt floor in household: frequency of households having dirt floors
17. hh owning land: frequency of households owning land
18. hh reducing meal portions in past 30 days: frequency of households who reduced meal portions in the past 30 days
19. average time to and from drinking water: mean amount of time to travel to and from nearest drinking water source per household
20. hh selling crops: frequency of households who sold crops in past 12 months
21. Amount of land owned per hh: mean amount of land owned per household

**Appendix B**

Summary of participatory workshop exercises

Watershed Map:　workshop participants were provided with a base satellite image of their local watershed, and asked to identify areas of increasing tree cover, decreasing tree cover, as well as important water sources

Change Matrix:　workshop participants identified important changes in the watershed in the most recent 3 year period; changes were weighted, and the top two changes were discussed in more detail according causes, consequences, and lessons learned

Worldview Analysis:　workshop participants identified key challenges facing the watershed. Priority challenges were selected through weighting, and then the 10 seed technique was used to analyze priority challenges according to level of community responsibility/influence

Peace and Reconciliation Analysis:　workshop participants shared accounts of peace and reconciliation in the watershed in the most recent 2 year period. Cases were selected from among all shared accounts and these were analysed in depth by the group

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
