# Peer review of "Community-Based Watershed Change: A Case Study in Eastern Congo"

_forests, doi:10.3390/f10060475_

Round 1
Reviewer 1 Report
General comments:
The manuscript presents an interesting evaluation of the impact of Village Savings and Loans Associations (SLVA) programs on the change in welfare in a watershed in South Kivu, Democratic Republic of Congo. The study was based on two sites (treated watershed in Kakumba and control watershed in Kambekulu), and the authors performed rigorous analysis approach using Theory of Change framework and impact evaluation methodology based on Difference in Differences (DID) approach. Explanation about the methodologies, however, was somehow lacking. The manuscript would be greatly benefited from detail explanations on: (1) the baseline biophysical and socioeconomic characteristics of the study area (treated and control), (2) types of data used and how they were collected or retrieved, (3) how the control watershed was selected, and (4) how the mixed method was applied for evaluating the impact of SLVA.
Specific comments:
Line 17: Please elaborate on how conflicts lead to forest loss.
Line 18: How does the peace process contribute to deforestation?
Line 21: Not clear, “requires nuanced understanding” of what?
Line 22: Needs elaborating on how the poor have been blamed for deforestation? Is it through the expansion of small scale agriculture? How does poverty associate with less deforestation?
Line 23: By whom the “many” refers to? International scientific communities? Governments?
Line 36: Need to elaborate on how the VSLA worked; would be useful for general readers who are not familiar with the context of Sub-Saharan Africa.
Line 47: Not sure what "to spread risk" means? Perhaps needs rephrasing.
Line 51: Please describe briefly about the study area in terms of demography, socioeconomic and environmental status.
Line 52: A brief description of what constitutes "Theory of Change" would be useful.
Line 56. A brief explanation about DID would be useful.
Line 58: What is the study period”
Line 58: What does the saving groups mean? Families under the VSLA scheme?
Line 64: Not clear what "treatment workshop participants" mean. Is this the "workshop participants from the treated SLVA group"?
Line 66: Again not sure what "comparison workshop participants" mean. Maybe "workshop participants from the control area"?
Line 69 and elsewhere: Here and elsewhere in the text, perhaps replace "comparison" with "control".
Table 2, row 1, column 1: Does this mean that improvement in indicators had occurred at a faster rate in the treated watershed compared to the control?
Table 2, row 2, column 1: Does this mean that the change in indicators had occurred at a similar rate both in treated and control watershed?
Table 2, row 3, column 1: Does this mean that the improvement in indicators had occurred at a slower rate in the treated watershed compared to the control? Or, the decline in indicators had occurred at a faster rate in the treated watershed compared to the control?
Line 125: The Materials and Methods section would be benefited from restructuring to improve clarity. Perhaps could subsequently describe: (1) the study area (in more detail) with some explanations about the biophysical conditions and the communities (ethnic background and socioeconomic status), then (2) data, and then (3) analysis steps.
Line 128: What were the considerations for selecting these sites? How did the control watershed selected?
Line 128: Does the control watershed had similar baseline characteristics as the treated watershed at the beginning of the survey period?
Line 130: What are the socioeconomic characteristics of the communities in the two watershed areas? What is the ethnic demography?
Line 135: What about the Kambekulu as control site? How did the Kambekulu selected? What is the methodology or considerations for selecting the control?
Line 156: Perhaps "using" rather than "and".
Line 160: Needs more explanation about the mixed methods applied.
Author Response
Response to Reviewer 1 comments Thank you for your helpful and insightful comments on the manuscript. We have made revisions accordingly. Please find those in blue font below. The manuscript presents an interesting evaluation of the impact of Village Savings and Loans Associations (VSLA) programs on the change in welfare in a watershed in South Kivu, Democratic Republic of Congo. The study was based on two sites (treated watershed in Kakumba and control watershed in Kambekulu), and the authors performed rigorous analysis approach using Theory of Change framework and impact evaluation methodology based on Difference in Differences (DID) approach. Explanation about the methodologies, however, was somehow lacking. The manuscript would be greatly benefited from detail explanations on: (1) the baseline biophysical and socioeconomic characteristics of the study area (treated and control), (2) types of data used and how they were collected or retrieved, (3) how the control watershed was selected, and (4) how the mixed method was applied for evaluating the impact of VSLA. Author note: All four of the above are addressed in the Materials and Methods section. Materials and Methods was previously placed between Discussion and Conclusion, according the the recommended Forests journal template. We have moved Materials and Methods so that it is now after Introduction and before Results. Specific comments: Line 17: Please elaborate on how conflicts lead to forest loss. Line 18: How does the peace process contribute to deforestation? Line 21: Not clear, “requires nuanced understanding” of what? Manuscript revisions: In a post-conflict situation such as the DRC, the peace process tends to focus on the logistical and institutional aspects of peace, such as negotiations between political groups, and reintegrating armed groups into civil society. This process typically does not take natural resource management into account and can contribute to environmental degradation as communities made desperate by conflict exploit forest resources [4]. The relationship of rural communities in the DRC with forests and natural resources is complex [5] and conflict in particular can affect wildlife and natural resources through multiple pathways [6] such as a direct impact on wildlife species, for example the mountain gorilla (Gorilla beringei) in Rwanda [7], to institutional impacts such as reducing enforcement effectiveness [8]. Addressing issues of conflict, peace, and deforestation requires understanding how complex pathways connect and interact Line 22: Needs elaborating on how the poor have been blamed for deforestation? Is it through the expansion of small scale agriculture? How does poverty associate with less deforestation? Manuscript revision: Worldwide, the poor have been blamed for deforestation, because of the expansion of small scale farming, the cutting and making of charcoal, and rotational farming (‘slash and burn’ agriculture) but poverty can often be associated with less deforestation [9] although causes of this association have not necessarily been elucidated. Line 23: By whom the “many” refers to? International scientific communities? Governments? Manuscript revision: An extensive review of the literature by Adams et al shows that there is a considerable body of evidence that biodiversity conservation needs to consider the factor of human poverty [10]. Line 36: Need to elaborate on how the VSLA worked; would be useful for general readers who are not familiar with the context of Sub-Saharan Africa. Manuscript revision (added to Results section): In the VSLA methodology, self-selected groups save regularly and make loans to group members, establishing their own bylaws, leadership, and loan interest rates. A high priority is placed on transparency which promotes confidence among members. At the end of a savings cycle (typically 12 months), group members will divide accumulated savings capital and interest earned from loans, each member receiving an amount proportionate to their investment. Most groups will agree to begin another savings cycle, and it is common for members to agree to invest a portion of their savings from the previous cycle, so that more capital is available for credit purposes. Growth of the savings amount as shown in Figure NNNN is in part a result of this group decision to reinvest in the subsequent cycle. Group members typically use both credit from the group as well as their accumulated savings to invest in small business, education, or diversify and protect their farms. Line 47: Not sure what "to spread risk" means? Perhaps needs rephrasing. Manuscript revision: One project in Kenya has shown that farmers involved in VSLA are better able to spread risk by making better decisions about climate-smart agriculture technologies such as crop diversification and agroforestry Line 51: Please describe briefly about the study area in terms of demography, socioeconomic and environmental status. Author note: the revision below is found in the Materials and Methods section, which has been moved up in the document so that it appears after the Introduction and before Results. Manuscript revision: The study area is located in South Kivu province of the DRC in Uvira territory. According to Demographic Health Survey (DHS) data from 2013 [42], in South Kivu 70.7 percent of women and 45.6 percent of men work in agriculture, 59.8 percent of women are literate, and 26 percent of children are underweight. Two watersheds in Uvira were selected for the study, Kakumba and Kambekulu and some basic characteristics of these watersheds are shown in table 3 below. Table 1: Characteristics of study watersheds Watershed Area (sq km) Population [43] % tree cover 2010 [44] Study designation Kakumba 37 13,917 22.6 treatment Kambekulu 54 16,338 27.5 comparison Figure 1: map of study watersheds Both watersheds are primarily agricultural, but have significant forest cover, especially in the upper watershed areas. The study site is adjacent to the Itombwe forest, a globally recognized area of biodiversity [45]. Line 52: A brief description of what constitutes "Theory of Change" would be useful. Manuscript revision: The TOC is intended to illustrate a logical progression from short term outputs, to long term impacts. Assumptions which underlie the pathway are included in Appendix C. In particular it is worth noting that ‘stakeholders are catalyzed and networked’ through formation of savings groups, and working with other local institutions such as churches, mosques, and schools; the expected outcome of this networking is that communities ‘value and protect their natural environment’. At each node in the TOC pathway, from outputs, through outcomes, to impact indicators were developed, and then tested using all available evidence from those indicators. A range of indicators and indicator formats were developed, including qualitative indicators used in participatory workshops, activity indicators collected on quarterly basis, quantitative indicators collected in a formal household survey, and remote sensing data in the form of NDVI. Testing of the TOC pathway using available indicators is further explained in the Discussion. Line 56. A brief explanation about DID would be useful. Manuscript revision (in Materials and Methods): Line 58: What is the study period” Manuscript revision (see also Materials and Methods section): Over the study period (Jan 2015 to May 2017), Line 58: What does the saving groups mean? Families under the VSLA scheme? Manuscript revision: 21 VSLA groups were established Line 64: Not clear what "treatment workshop participants" mean. Is this the "workshop participants from the treated SLVA group"? Manuscript revision: Workshop participants in the treatment watershed, where VSLA groups were established, identified increased availability of capital as a result of savings group activity as one of the most significant changes in the watershed, while the workshop participants in the comparison watershed, where no VSLA groups were established, identified a lack of capital and increasing poverty as a significant change over the study period. Line 66: Again not sure what "comparison workshop participants" mean. Maybe "workshop participants from the control area"? See above Line 69 and elsewhere: Here and elsewhere in the text, perhaps replace "comparison" with "control". Done
Difference in differences analysis compares the change in condition of two groups, the treatment and comparison (control) over a period of time--essentially a statistical ‘before and after’ assessment.

Reviewer 2 Report
This manuscript presents results of a relatively interesting study of the short-term impacts of community interventions in financial management, sustainable farming and environmental awareness in a region in the eastern Congo. While the manuscript is an interesting case study, in its current form it is not suitable for publication. The study is not framed in the context of the broader literature on community development and environmental interventions and the study design and the methodology are not sufficiently rigorous to support conclusions. The authors might consider presenting the study for publication in a regional journal, or repeating some of the analysis in a few years, when the results are more definitive.
The structure of the presentation is confusing. The authors should stick to the standard scientific presentation order: introduction, methods, results, discussion and conclusions.
Introduction
To justify publication in an international journal, the authors need to clearly frame the study and the objectives in the literature on community development and forest management and protection. What is the area or discipline of interest for the study, what are some of the general findings and conceptual models used in this area and what is the gap in knowledge that this study is aiming to address? The authors should review the broader principles of community forest management, community development and the relationships between forests and poverty.
Methods
The design of the methodology is not that clear. There seems to be an attempt to undertake a Before-After-Control-Impact (BACI) design but it is not clear if the community interventions (treatments) were undertaken in both watersheds, or only one. The study refers to the elements of a theory of change, but it is not clear how the elements of this theory were derived in this context. The theory seems rather simple given the complex social-ecological system in which the interventions are being introduced.
The analysis is also not clear. Did the analysis compare the baseline indicator data against the control catchment, or after two years of ‘treatment’? How was the difference in difference analysis applied, at a household or catchment level?
Results
The results in table 2 need more extensive presentation, indicating the values, statistical differences and nature of the difference for each indicator.
Discussion
The manuscript needs to discuss the results in the context of the wider literature, as indicated above, and discuss the gap in knowledge more specifically being addressed in this study.
Author Response
Response to Reviewer 2 comments Thank you for your review. We have changed the order of the paper, so that Materials and Methods appears after introduction. We have tried to expand both Materials and Methods as well as Introduction to better contextualize this work. Please see attached manuscript revision. This manuscript presents results of a relatively interesting study of the short-term impacts of community interventions in financial management, sustainable farming and environmental awareness in a region in the eastern Congo. While the manuscript is an interesting case study, in its current form it is not suitable for publication. The study is not framed in the context of the broader literature on community development and environmental interventions and the study design and the methodology are not sufficiently rigorous to support conclusions. The authors might consider presenting the study for publication in a regional journal, or repeating some of the analysis in a few years, when the results are more definitive. The structure of the presentation is confusing. The authors should stick to the standard scientific presentation order: introduction, methods, results, discussion and conclusions. Introduction To justify publication in an international journal, the authors need to clearly frame the study and the objectives in the literature on community development and forest management and protection. What is the area or discipline of interest for the study, what are some of the general findings and conceptual models used in this area and what is the gap in knowledge that this study is aiming to address? The authors should review the broader principles of community forest management, community development and the relationships between forests and poverty. Methods The design of the methodology is not that clear. There seems to be an attempt to undertake a Before-After-Control-Impact (BACI) design but it is not clear if the community interventions (treatments) were undertaken in both watersheds, or only one. The study refers to the elements of a theory of change, but it is not clear how the elements of this theory were derived in this context. The theory seems rather simple given the complex social-ecological system in which the interventions are being introduced. The analysis is also not clear. Did the analysis compare the baseline indicator data against the control catchment, or after two years of ‘treatment’? How was the difference in difference analysis applied, at a household or catchment level? Results The results in table 2 need more extensive presentation, indicating the values, statistical differences and nature of the difference for each indicator. Discussion The manuscript needs to discuss the results in the context of the wider literature, as indicated above, and discuss the gap in knowledge more specifically being addressed in this study.

Reviewer 3 Report
This sort of research should be applauded in its scope and its aims. Nature-society interrelationships are indeed complex and the urgency of needing to improve the trajectories of sustainable development in the areas discussed in this paper is high. However, I find some shortcomings that must first be addressed, before it would be suitable for publishing. I recommend reconsidering this article after major revisions.
First and foremost:
The entire argument of the paper appears tautological– that if you intervene to improve the system by doing X & Y (say establishing tree planting groups and savings groups) and you measure those inputs as outputs (measuring trees planted and numbers of savings groups), you see improvements in X &Y. So what is the finding?
This can relate to the ToC framework (addressed more specifically below). A robust ToC framework is explicit about the assumptions (along the whole change logic pathway, from the problem identification, to interventions & outcomes to the goals) (see Sayer et al., 2016). Perhaps the paper could be more streamlined by restructuring the delivery of its story: an assessment of a TOC in delivering conservation and development benefits in eastern DRC. As written, the science experimentation and the TOC are not carefully, or obviously woven together.
I sympathize with the difficulties in measuring the effectiveness of programs, I have been doing this sort of impact-oriented research in lots of different contexts.
The authors should be weary of presenting success stories when there are most certainly lots of endogenous characteristics of each landscape affecting the outcomes of interventions. It does not necessarily discount your story – but all those landscape contexts should be made more transparent and clearer to the reader. The effect of the intervention while looking good in terms of short-term outputs, is not adequately framed in terms of long-term impacts – address the limitations head on. The final statements about ‘costs vs benefits’ is a good one but not based in any presentation of the ‘costs’ so is impossible to verify or validate as a reader.
As another general statement: as a researcher heavily invested in deriving criteria and indicators for forest landscape restoration, I am concerned that the metrics used for success here do not substantively inform us of what management practices are good for who… More attention to the limitations of short term-metrics (like number of trees planted, or workshops attended), and how they might be refined to reflect robust societal learning would greatly improve the paper.
A structural flaw – The results are presented before the methods, making it quite difficult to read. The reader does not have the context of the two watersheds before the authors start presenting the results of a treatment (which readers aren’t aware of until they get to it later).
Specific Comments
Line 19: what peace process (do not assume readers know the context peace-building in the DRC – this could be added into a much more context heavy introduction)
Line 21: whilst I agree, this is abstract and vague – elaborate and ground that statement in more literature and tangible reality.
Line 22-24: Maybe what makes them poor is that they are unable to use those assets to get ahead in any markets. These are statements that don’t capture the enormity of conservation-development tradeoff literature behind them. Please add more context, perhaps to the local contexts. For instance, you write “It is recognized by many that biodiversity conservation needs to consider the factor of human poverty.” What about all other dimensions of human development? Exclusive development via green-grabbing? It is also recognized by many that conservation will not succeed if people are allowed to use their local resources for development (see Terborgh and Peres (2017) * I’m trying to make the case that you should assume an audience reads statements from diverse perspectives of what exactly is the problem. As it stands this is too general of a statement that should be followed up by some elaboration to make it clearer of what the problem is, and according to who.
Line 27-28: Be clearer: community networks do what for forest management? Under what conditions are they affective? (From Ostrom onwards, lots written about local management to name a few: (De Royer et al., 2018, Gilmour, 2016, Berkes, 2004))
Line 31: “social networks have been shown to be closely tied to building agroecological resilience”. Clarify: what about social networks ties to agroecological resilience? Social networks exist regardless of agroecological resilience. Something about the network has to influence either positively or negatively to agroecological resilience. For a more insightful phrase: “networks that are characterized by XYZ are more positively linked to agroecological resilience”
Line 34: While this is true it is a one-sided, uncritical statement. Participation is costly to poor people who have other things to do (to anyone really). All of these endeavors have transactions costs that might exceed the opportunity cost to not participate. Participation can become tyrannical (Cooke and Kothari, 2001); and subsequent critical literature on participation in NRM (Langston et al., 2019, Cleaver, 2002, Gaynor, 2013, Fleishman, 2009). The costs of participation in programs like the ones presented in this paper can be prohibitive or can lead to entrenched inequality. This relates to the whole summary of the paper (as evident in the last sentence of the conclusion): there is no explicit or critical presentation of the costs of establishing community endeavors like the ones presented in the paper. A critical presentation on why this is difficult, what costs are, how these initiatives might be limited in their ability to deliver transformative development opportunities (more than say a marginal alleviation of poverty) would add some oomph to the paper.
Line 37: “Extensive study of VSLA methodology in Sub-Saharan Africa can have a positive effect on poverty although outcomes vary from study to study”. Clarify: the study of methodology affects poverty? Or do you mean many studies show how VSLA can alleviate poverty? Fix the grammar to reflect your meaning.
Lines 39-41: Authors identify conflicting results. This is a chance to be critical of RCTs to inform us of lessons from interventions in complex social-ecological systems (as the watersheds analyzed in this paper most certainly are). Perhaps the lessons of one intervention cannot be transferred from place to place (much of the place-based transdisciplinary science literature expresses this)
Line 44: “A study specific to DRC shows that VSLA can contribute to building capacity of local institutions in mining regions”. Not very helpful statement. Can say VSLA can contribute - but in the contexts of XYZ, and the leverage points for positive outcomes might be A,B,C) If it is the local institutional arrangements and their effectiveness to start with then we have a problem of causality.
Line 49: unclear – who advocates for complementary initiatives to reduce household vulnerability?
Line 52: TOC framework appears. Describe how this is relevant and contextualize ToC to the literature / norms of organizational strategies (as mentioned above (Sayer et al., 2016))
Line 53: “…based on a Theory of Change framework which integrates VSLA as a platform as well as existing institutions…” unclear: the TOC integrates the VSLA as a platform? What does this mean - can you explain?
Results section: reader has a hard time understanding what the results are without understanding what the researchers did first. Propose putting methods before the results. Otherwise there is no logical narrative to the paper.
Line 80: Spell out NDVI the first time in the text.
Figures and Tables: Add comprehensively self-contained figure legends. What is a cycle? What is CD?
For table 2: Hard to read a list of 15 indicators in one block of text. Suggest different table style - present a list of all indicators along a leftmost column, with three more columns - either Improvement, no improvement or decline -tick the cell. order list of indicators from (top improvements to bottom declines)
Discussion: what is the increasing saving coming from? What behaviours are changing? What economic opportunities have changed to allow for savings?
Lines 108-111: would like to know what the criteria are that underpin the indicators. This should be presented more thoroughly in the results section.
Figure 6: Suggest not to present part of a TOC before you present the whole ToC. It does not add readability or logic to the text (TOCs are meant to convey logic afterall)
Methods: Confused about the placement of methods after the discussion and before the conclusion. It tells the narrative in a backwards way (rather: set up the scene with intro, describe your plot with the methods, convey the climax of the story with the results/discussion, then wrap up with conclusion).
Lines 150-161: Where is it laid out how data was collected for (1) treatment and (2) control? (all that is evident is an aggregate there… (authors identify how many interviews were conducted in total at the beginning and end, not in each watershed sample) Also, why these sample sizes? This does not express a process that could be replicable in its current form.
Lines 157: “In addition, treatment communities reported every 3 months on metrics such as trees planted, and savings group activities” – is this the data that the study is based on? I would expect a discussion on the limitations of self-reporting by treatment communities that might indeed have a vested interest in expressing success….
Figure 8: need more comprehensive description of this ToC. Is this a subset of a bigger ToC? where have you identified the assumptions built into your ToC? A robust ToC should be about How, not just What.
Line 178: again – what discussion of costs allow the reader to evaluate whether the costs of the project outweighed the benefits?
BERKES, F. 2004. Rethinking community‐based conservation. Conservation biology, 18, 621-630.
CLEAVER, F. 2002. Reinventing institutions: Bricolage and the social embeddedness of natural resource management. The European journal of development research, 14, 11-30.
COOKE, B. & KOTHARI, U. 2001. Participation: The new tyranny?, Zed books.
DE ROYER, S., VAN NOORDWIJK, M. & ROSHETKO, J. 2018. Does community-based forest management in Indonesia devolve social justice or social costs? International Forestry Review, 20, 167-180.
FLEISHMAN, R. 2009. To participate or not to participate? Incentives and obstacles for collaboration. The collaborative public manager, 31-52.
GAYNOR, N. 2013. The tyranny of participation revisited: international support to local governance in Burundi. Community Development Journal, 49, 295-310.
GILMOUR, D. 2016. Forty years of community-based forestry: A review of its extent and effectiveness. FAO Forestry Paper, 176.
LANGSTON, et al. 2019. Science Embedded in Local Forest Landscape Management Improves Benefit Flows to Society. Frontiers in Forests and Global Change, 2.
SAYER, J., et al. 2016. Measuring the effectiveness of landscape approaches to conservation and development. Sustainability Science, 1-12.
TERBORGH, J. & PERES, C. A. 2017. Do Community-Managed Forests Work? A Biodiversity Perspective. Land, 6, 22.
Author Response
Response to Reviewer 3 comments Community based watershed management: a case study in Eastern Congo Thank you for your thorough and thoughtful comments. We have tried to address each concern in the manuscript. Please find our edits and additional comments in blue font below. This sort of research should be applauded in its scope and its aims. Nature-society interrelationships are indeed complex and the urgency of needing to improve the trajectories of sustainable development in the areas discussed in this paper is high. However, I find some shortcomings that must first be addressed, before it would be suitable for publishing. I recommend reconsidering this article after major revisions. First and foremost: The entire argument of the paper appears tautological– that if you intervene to improve the system by doing X & Y (say establishing tree planting groups and savings groups) and you measure those inputs as outputs (measuring trees planted and numbers of savings groups), you see improvements in X &Y. So what is the finding? This can relate to the ToC framework (addressed more specifically below). A robust ToC framework is explicit about the assumptions (along the whole change logic pathway, from the problem identification, to interventions & outcomes to the goals) (see Sayer et al., 2016). Perhaps the paper could be more streamlined by restructuring the delivery of its story: an assessment of a TOC in delivering conservation and development benefits in eastern DRC. As written, the science experimentation and the TOC are not carefully, or obviously woven together. I sympathize with the difficulties in measuring the effectiveness of programs, I have been doing this sort of impact-oriented research in lots of different contexts. The authors should be weary of presenting success stories when there are most certainly lots of endogenous characteristics of each landscape affecting the outcomes of interventions. It does not necessarily discount your story – but all those landscape contexts should be made more transparent and clearer to the reader. The effect of the intervention while looking good in terms of short-term outputs, is not adequately framed in terms of long-term impacts – address the limitations head on. The final statements about ‘costs vs benefits’ is a good one but not based in any presentation of the ‘costs’ so is impossible to verify or validate as a reader. As another general statement: as a researcher heavily invested in deriving criteria and indicators for forest landscape restoration, I am concerned that the metrics used for success here do not substantively inform us of what management practices are good for who… More attention to the limitations of short term-metrics (like number of trees planted, or workshops attended), and how they might be refined to reflect robust societal learning would greatly improve the paper. A structural flaw – The results are presented before the methods, making it quite difficult to read. The reader does not have the context of the two watersheds before the authors start presenting the results of a treatment (which readers aren’t aware of until they get to it later). Author comment: By restructuring the manuscript and modifying content based on reviewer input, we hope we have made a tighter connection between activities, outputs, outcomes, and impact. We have tried to point out that the project itself really only provides one primary input, training. All other actions, tree planting, community forest managment, enforcement of environmental bylaws, savings, are voluntary actions taken by the communities themselves. Specific Comments Line 19: what peace process (do not assume readers know the context peace-building in the DRC – this could be added into a much more context heavy introduction) Manuscript revision: In a post-conflict situation such as the DRC, the peace process tends to focus on the logistical and institutional aspects of peace, such as negotiations between political groups, and reintegrating armed groups into civil society. This process typically does not take natural resource management into account and can contribute to environmental degradation as communities made desperate by conflict exploit forest resources Line 21: whilst I agree, this is abstract and vague – elaborate and ground that statement in more literature and tangible reality. Manuscript revision: The relationship of rural communities in the DRC with forests and natural resources is complex [5] and conflict in particular can affect wildlife and natural resources through multiple pathways [6] such as a direct impact on wildlife species, for example the mountain gorilla (Gorilla beringei) in Rwanda [7], to institutional impacts such as reducing enforcement effectiveness [8]. Addressing issues of conflict, peace, and deforestation requires understanding how complex pathways connect and interact Line 22-24: Maybe what makes them poor is that they are unable to use those assets to get ahead in any markets. These are statements that don’t capture the enormity of conservation-development tradeoff literature behind them. Please add more context, perhaps to the local contexts. For instance, you write “It is recognized by many that biodiversity conservation needs to consider the factor of human poverty.” What about all other dimensions of human development? Exclusive development via green-grabbing? It is also recognized by many that conservation will not succeed if people are allowed to use their local resources for development (see Terborgh and Peres (2017) * I’m trying to make the case that you should assume an audience reads statements from diverse perspectives of what exactly is the problem. As it stands this is too general of a statement that should be followed up by some elaboration to make it clearer of what the problem is, and according to who. Manuscript revision: Nevertheless, community-based approaches to conservation are controversial and have shown mixed results [10–12]. Berkes asserts that it is important to consider cases where community based approaches have worked and where they haven’t worked [11]. Line 27-28: Be clearer: community networks do what for forest management? Under what conditions are they affective? (From Ostrom onwards, lots written about local management to name a few: (De Royer et al., 2018, Gilmour, 2016, Berkes, 2004)) Manuscript revision: Based on the authors’ experience and summarized in a case study, community networks have been demonstrated to be an effective way of managing forests at the watershed level in northern Thailand [17,18]. In that particular case study, the local network allowed communities to pool resources, make maps and forest management plans, share local knowledge, and build better relationships with government authorities. Reviews of the literature point to a considerable body of evidence discussing how community networks and local involvement can enhance forest management by devolving the decision making process, integrating local knowledge, and increasing secure tenure [11,15,19]. Line 31: “social networks have been shown to be closely tied to building agroecological resilience”. Clarify: what about social networks ties to agroecological resilience? Social networks exist regardless of agroecological resilience. Something about the network has to influence either positively or negatively to agroecological resilience. For a more insightful phrase: “networks that are characterized by XYZ are more positively linked to agroecological resilience” Manuscript revision: On the farming side of the equation, social networks allow sharing of local germplasm, sharing knowledge of adaptive farming strategies, and the rapid scaling up of resilient techniques, and have been shown to be closely tied to building agroecological resilience [20]. Line 34: While this is true it is a one-sided, uncritical statement. Participation is costly to poor people who have other things to do (to anyone really). All of these endeavors have transactions costs that might exceed the opportunity cost to not participate. Participation can become tyrannical (Cooke and Kothari, 2001); and subsequent critical literature on participation in NRM (Langston et al., 2019, Cleaver, 2002, Gaynor, 2013, Fleishman, 2009). The costs of participation in programs like the ones presented in this paper can be prohibitive or can lead to entrenched inequality. This relates to the whole summary of the paper (as evident in the last sentence of the conclusion): there is no explicit or critical presentation of the costs of establishing community endeavors like the ones presented in the paper. A critical presentation on why this is difficult, what costs are, how these initiatives might be limited in their ability to deliver transformative development opportunities (more than say a marginal alleviation of poverty) would add some oomph to the paper. Author comments: Reviewer points are fair, and we also have concerns about time and cost for farmers. We do have data on costs, both to farmers and the project. Can we consider a separate publication on this? We feel that treating this adequately in the current manuscript would result in a paper that tries to address too many themes at once. Line 37: “Extensive study of VSLA methodology in Sub-Saharan Africa can have a positive effect on poverty although outcomes vary from study to study”. Clarify: the study of methodology affects poverty? Or do you mean many studies show how VSLA can alleviate poverty? Fix the grammar to reflect your meaning. Manuscript revision: Extensive study in Sub-Saharan Africa has shown that the VSLA methodology can have a positive effect on poverty Lines 39-41: Authors identify conflicting results. This is a chance to be critical of RCTs to inform us of lessons from interventions in complex social-ecological systems (as the watersheds analyzed in this paper most certainly are). Perhaps the lessons of one intervention cannot be transferred from place to place (much of the place-based transdisciplinary science literature expresses this) Manuscript revision: These varying, and sometimes conflicting results underscore the caution that should be exercised when trying to extrapolate lessons learned from complex environments including the current case study. Line 44: “A study specific to DRC shows that VSLA can contribute to building capacity of local institutions in mining regions”. Not very helpful statement. Can say VSLA can contribute - but in the contexts of XYZ, and the leverage points for positive outcomes might be A,B,C) If it is the local institutional arrangements and their effectiveness to start with then we have a problem of causality. Manuscript revision: A study specific to DRC shows that VSLA can contribute improving financial management skills and financial transparency of local institutions in mining regions which has helped address issues of corruption Line 49: unclear – who advocates for complementary initiatives to reduce household vulnerability? Manuscript revision: A study in Zimbabwe shows that savings groups have a positive effect on financial services in rural areas, but study authors advocate for complementary initiatives to reduce household vulnerability Line 52: TOC framework appears. Describe how this is relevant and contextualize ToC to the literature / norms of organizational strategies (as mentioned above (Sayer et al., 2016)) Manuscript revision: A conceptual framework, such as a TOC, provides a way to articulate goals, assumptions, and metrics [40]. In a complex context, such as a watershed or landscape level community effort, having such a framework with which to iteratively assess progress towards short, mid, and long term goals becomes critical Line 53: “…based on a Theory of Change framework which integrates VSLA as a platform as well as existing institutions…” unclear: the TOC integrates the VSLA as a platform? What does this mean - can you explain? Manuscript revision: This paper reports on a community-based watershed initiative in Uvira Territory of South Kivu, in eastern Democratic Republic of the Congo. A Theory of Change (TOC) framework outlines a watershed level development model which integrates VSLA as a platform as well as existing institutions such as churches, mosques, schools, for strengthening household economic condition and catalyzing downstream impacts such as environmental restoration. Results section: reader has a hard time understanding what the results are without understanding what the researchers did first. Propose putting methods before the results. Otherwise there is no logical narrative to the paper. Done. We were also uncomfortable with this format which was prescribed in the Forests template. Much more used to the standard, Introduction-Materials & Methods-Results format Line 80: Spell out NDVI the first time in the text. Done Figures and Tables: Add comprehensively self-contained figure legends. What is a cycle? What is CD? Done For table 2: Hard to read a list of 15 indicators in one block of text. Suggest different table style - present a list of all indicators along a leftmost column, with three more columns - either Improvement, no improvement or decline -tick the cell. order list of indicators from (top improvements to bottom declines) Done Discussion: what is the increasing saving coming from? What behaviours are changing? What economic opportunities have changed to allow for savings? Manuscript revision: In the VSLA methodology, self-selected groups save regularly and make loans to group members, establishing their own bylaws, leadership, and loan interest rates. A high priority is placed on transparency which promotes confidence among members. At the end of a savings cycle (typically 12 months), group members will divide accumulated savings capital and interest earned from loans, each member receiving an amount proportionate to their investment. Most groups will agree to begin another savings cycle, and it is common for members to agree to invest a portion of their savings from the previous cycle, so that more capital is available for credit purposes. Growth of the savings amount as shown in Figure NNNN is in part a result of this group decision to reinvest in the subsequent cycle. Group members typically use both credit from the group as well as their accumulated savings to invest in small business, education, or diversify and protect their farms. Lines 108-111: would like to know what the criteria are that underpin the indicators. This should be presented more thoroughly in the results section. Manuscript revision: The TOC is intended to illustrate a logical progression from short term outputs, to long term impacts. In particular it is worth noting that ‘stakeholders are catalyzed and networked’ through formation of savings groups, and working with other local institutions such as churches, mosques, and schools; the expected outcome of this networking is that communities ‘value and protect their natural environment’. At each node in the TOC pathway, from outputs, through outcomes, to impact indicators were developed, and then tested using all available evidence from those indicators. A range of indicators and indicator formats were developed, including qualitative indicators used in participatory workshops, activity indicators collected on quarterly basis, quantitative indicators collected in a formal household survey, and remote sensing data in the form of NDVI. Testing of the TOC pathway using available indicators is further explained in the Discussion. Figure 6: Suggest not to present part of a TOC before you present the whole ToC. It does not add readability or logic to the text (TOCs are meant to convey logic afterall) Done. changing order of Materials & Methods addresses this Methods: Confused about the placement of methods after the discussion and before the conclusion. It tells the narrative in a backwards way (rather: set up the scene with intro, describe your plot with the methods, convey the climax of the story with the results/discussion, then wrap up with conclusion). Materials & Methods has now been placed before Results Lines 150-161: Where is it laid out how data was collected for (1) treatment and (2) control? (all that is evident is an aggregate there… (authors identify how many interviews were conducted in total at the beginning and end, not in each watershed sample) Also, why these sample sizes? This does not express a process that could be replicable in its current form. Manuscript revision: A baseline study was conducted in Jan 2015 using participatory workshops and a household survey of 96 randomly selected households from Kakumba watershed and 87 randomly selected households from Kambekulu watershed. A similar study was conducted at the end of the pilot project, in May 2017 also using participatory workshops (see Appendix B) and a household survey of 160 randomly selected households from each watershed for a total of 320 households sampled (see Appendix A for more details on indicators collected). Total sample size was based on the total population in the watersheds with the aim to obtain a confidence interval of 7.5 percent on estimates of the mean. The 2017 sample size was increased so that a similar level of confidence could be obtained when data was disaggregated by factors such as watershed, gender, participation. Lines 157: “In addition, treatment communities reported every 3 months on metrics such as trees planted, and savings group activities” – is this the data that the study is based on? I would expect a discussion on the limitations of self-reporting by treatment communities that might indeed have a vested interest in expressing success…. Manuscript revision: These short term metrics were used to corroborate the longer term metrics measured in the household survey and participatory workshops. All metrics were examined to test the Theory of Change at each node from output to impact level using a contribution analysis approach Figure 8: need more comprehensive description of this ToC. Is this a subset of a bigger ToC? where have you identified the assumptions built into your ToC? A robust ToC should be about How, not just What. Manuscript revision: The TOC is intended to illustrate a logical progression from short term outputs, to long term impacts. Assumptions which underlie the pathway are included in Appendix C. In particular it is worth noting that ‘stakeholders are catalyzed and networked’ through formation of savings groups, and working with other local institutions such as churches, mosques, and schools; the expected outcome of this networking is that communities ‘value and protect their natural environment’. At each node in the TOC pathway, from outputs, through outcomes, to impact indicators were developed, and then tested using all available evidence from those indicators. A range of indicators and indicator formats were developed, including qualitative indicators used in participatory workshops, activity indicators collected on quarterly basis, quantitative indicators collected in a formal household survey, and remote sensing data in the form of NDVI. Testing of the TOC pathway using available indicators is further explained in the Discussion. Line 178: again – what discussion of costs allow the reader to evaluate whether the costs of the project outweighed the benefits? Author comments: A fair point. We do have data on project costs, but including a discussion of costs directly seems like it would extend the scope of the paper. Revision below removes remark about cost which admittedly was not substantiated in the draft as currently written. Manuscript revision: They support the results of other studies that show that catalyzing community involvement can be an effective alternative for promoting the improvement of watershed/ecosystem health and forest management BERKES, F. 2004. Rethinking community‐based conservation. Conservation biology, 18, 621-630. CLEAVER, F. 2002. Reinventing institutions: Bricolage and the social embeddedness of natural resource management. The European journal of development research, 14, 11-30. COOKE, B. & KOTHARI, U. 2001. Participation: The new tyranny?, Zed books. DE ROYER, S., VAN NOORDWIJK, M. & ROSHETKO, J. 2018. Does community-based forest management in Indonesia devolve social justice or social costs? International Forestry Review, 20, 167-180. FLEISHMAN, R. 2009. To participate or not to participate? Incentives and obstacles for collaboration. The collaborative public manager, 31-52. GAYNOR, N. 2013. The tyranny of participation revisited: international support to local governance in Burundi. Community Development Journal, 49, 295-310. GILMOUR, D. 2016. Forty years of community-based forestry: A review of its extent and effectiveness. FAO Forestry Paper, 176. LANGSTON, et al. 2019. Science Embedded in Local Forest Landscape Management Improves Benefit Flows to Society. Frontiers in Forests and Global Change, 2. SAYER, J., et al. 2016. Measuring the effectiveness of landscape approaches to conservation and development. Sustainability Science, 1-12. TERBORGH, J. & PERES, C. A. 2017. Do Community-Managed Forests Work? A Biodiversity Perspective. Land, 6, 22.

Round 2
Reviewer 1 Report
The manuscript is much improved in terms of clarity. I am pleased about the changes made by the authors, and recommend the manuscript to be published.
Author Response
Thank you for your helpful comments through this process.
Please find attached a PDF version of the manuscript draft with new changes in blue font. We hope that the new revisions tie the paper together more.

Reviewer 2 Report
The authors have addressed some of my previous concerns about the context, methods and presentation of results. However, there a three major problems that, in my view, prevent acceptance of the paper. Overall, the manuscript continues to read like an evaluation report for a development project rather than a properly conducted research study. While the outcomes of the project generally appear to be positive, there is insufficient evidence to suggest that the interventions in the project have had a significant impact on forest resources, or that there is a clear causal relationship between the interventions and change in ecosystem condition. Therefore, the findings and conclusions are not support by the results of the study.
The causal relationship between autonomous savings groups, community leadership, community forest management and improved forest condition is not clear. The manuscript reports primarily on social outcomes associated with the development of savings groups. These are laudable but are not really relevant to a study in Forests journal. The primary positive outcome for forests seems to be 'number of trees planted per household'. It is not clear that this is due to support to develop the savings groups, or other activities to promote environmental restoration. If it is the latter, then why the focus on savings groups? The outcomes for other measures of forest, ecosystem or watershed condition are inconclusive. primarily due to the short term (2 years) of the study. This is to be expected, social interventions are unlikely to lead to detectable changes in ecosystem condition in this time frame. However, the authors cannot claim there is a 'Positive impact is observed in the treatment watershed on ecosystem health'.
Secondly, the analysis and results are still confusing. How were the 96 and 87 households that were surveyed in the study 'randomly' selected? Was the analysis conducted on simple averages of the indicators across these households? The mean and SE for the indicators in the two watersheds at the two points in time are not presented.
The manuscript does not discuss the mechanisms for change and how these compare with other similar studies.
Author Response
Thanks for reviewing this draft and for your patience with this process. We have made new revisions which you will find in the attached PDF. All new changes are in blue font. We hope that these changes will address your concerns, in particular linking this particular study to the existing literature.
All the best.

Reviewer 3 Report
Thank you for your revisions and comments. The manuscript is improved, and there are now minor revisions to be made. First about impacts, second about the methodological transparency.
Careful with terms ‘impact’, ‘outcomes’. There is a liberal use of ‘impact’ and this should be reconsidered where used throughout the text. Consider these generally accepted definitions
Process/implementation evaluation (short term) determines whether activities have been implemented as intended.
Outcome/effectiveness evaluation (short to medium term) assesses a target population by assessing the progress in the outcomes or outcome objectives that a program aims to achieve.
Impact evaluation (longer term) assesses effectiveness according to some ultimate goals.
As such:
Line 9 (abstract): Positive impact is observed in the treatment watershed….: Be careful to not extrapolate yet to impact. Outcomes are not Impacts. Impacts are longer term shifts towards a goal, and you are duly more cautious about claiming impact in your conclusions while acknowledging your positive outcomes or influences.
Line 10 (abstract): “Results also suggest an impact on peace conditions which, while fragile, offer hope for continued restorative action by communities.” What kind of impact? (or is Impact the right word..) Be more precise. Something like – Results of our training led to XYZ in the community, which suggests a positive influence on the peace conditions.” (you are clearer about this in the results section)(also lines 211-216 to help here)
A flag raised in the methods:
Line 99 (Methods): your treatment watershed was chosen randomly? This does not sound credible. Why are the authors, (if they are part of the team) working in these watersheds and what contexts led you to them. These reasons must be made explicit. It is fine to purposively work in some area and consider a neighboring area for comparative purposes, but as written it looks arbitrarily objective… while conservation/development work is hardly arbitrary nor objective. Why was one chosen as treatment and one as control. A more constructive description of what the authors have done to determine where they are working is required. This comes out of concerns in the literature that action researchers/development & conservation interveners must be more reflective and honest about their positions, why they are doing what they are doing. This brings up a question of style in the writing. Are the authors the ones who have designed the project and designed the assessment study? this should come across in the style of writing. see below.
For example in lines: "From 2015 to 2017, a community-based watershed project was implemented in the study area through a collaboration between two organizations: Ebenezer Ministries International (EMI), the local implementer and Plant With Purpose [46], the international partner." The authors in their language style throughout the text have distanced themselves from the project, and the study. Are the authors not part of a team that did the project? and are they assessing their own project? this is all fine but good accountability in science is necessary. Would like to see more transparency in who is writing the paper, doing the project, assessing the project. This requires more active, first person writing.
As such - in the abstract : "(insert word)Study looks at a community-based environmental restoration project based on a framework (Theory of Change) which networks communities through autonomous savings groups, churches, mosques, schools and a community leadership network with the goal of catalyzing sustainable farming, reforestation, and community forest management". Would prefer 'we, (the authors), have examined our project....", "our methods include XYZ".
There are too many instances of not linking the study agents with the project agents, and the authors with either one. So this needs reconfiguring throughout the text.
comment:
Line 174: clear that you are talking about outcomes here (as laid out in TOC) – appreciate this sort of precision in language and could be used to better hedge your trajectory towards impact later in the text.
Author Response
Thank you for your thorough and patient input.
Please find attached a draft of the manuscript with new revisions in blue font. To your comments, we have tried to use more cautious language related to impact and have tried to tie the paper together more tightly.
